# The Mealtime Behavior Problems of Children with Developmental Disabilities and the Teacher’s Stress in Inclusive Preschools

**DOI:** 10.3390/children10030441

**Published:** 2023-02-24

**Authors:** Chen-Ya Juan

**Affiliations:** Center of Teacher Education, Minghsin University of Science and Technology, Hsinchu 30401, Taiwan; chenyajuan@must.edu.tw

**Keywords:** mealtime behavior problems, children with developmental disabilities, inclusive preschool, preschool teacher’s stress

## Abstract

With an increasing number of children with developmental disabilities entering inclusive preschools, preschool teachers face more behavioral problems in class. Preschool teachers typically attempt to address mealtime behavior problems of children with and without developmental disabilities simultaneously in class. This study used qualitative research to identify the stress triggers of preschool teachers addressing the mealtime behavior problems of children with developmental disabilities. Five preschool teachers attended semi-structured interviews. The results indicated that most children with developmental disabilities had problems with eating only preferred foods, using eating utensils appropriately during mealtime, becoming distracted from eating, and becoming frustrated with the classroom routine. Although solving these problems triggered stress in the preschool teachers, their stress was mainly in response to the children’s parents, other children’s imitation of inappropriate mealtime behaviors, and classroom schedule time management. Most of the preschool teachers stated that they had insufficient support. Preschool teachers require specialized information and strategies for improving the mealtime behaviors of children with developmental disabilities.

## 1. Introduction

Many children with special needs have behavior-related eating problems, including having particular food preferences, exhibiting aggression during mealtime, and becoming distracted from eating [1,2]. Many exhibit behavior problems during mealtime, including eating only small quantities of food, spitting out food, eating too quickly without chewing, eating too slowly without swallowing; chewing insufficiently, exhibiting rumination syndrome, pushing away food, and demanding specific feeders, specific food settings, or food with particular textures [3]. These inappropriate mealtime behaviors (IMBs) impede children from receiving the necessary nourishment from food, which may cause serious health problems or developmental obstacles [4].

Mealtime behavior problems are one of preschool teachers’ most common daily challenges. According to my on-site observations, the stress that preschools teacher experience in relation to children’s mealtime behavior problems typically results from the children’s parents, time pressure, other children’s imitation of misbehaviors, and the preschool administrator’s concern. Some parents in Asia have a tendency to be overinvolved in their children’s eating in preschool out of concern that their children are receiving insufficient food and nutrition to support them developmentally. Parents might review the school menu monthly and request that the teacher feed their children during mealtime. Instead of teaching their children appropriate mealtime behaviors at home, parents believe that teachers must become more efficient and trained in improving their children’s mealtime behaviors. Laws and regulations in Taiwan also address the importance of nutrition for children with disabilities [5,6]. Many children with developmental disabilities demonstrate IMBs in preschools, which may result in insufficient nutritional intake. Some children wanted to watch cartoons or play with mobile devices while eating, to wait for teachers to feed them, or to run around and eat in the classroom, and others were unable to spoon-feed themselves. These problem behaviors increase teachers’ stress levels and directly impede the children’s food intake.

With 1 to 15 children per class, managing all children’s mealtime behaviors is challenging for teachers. Before mealtime, the teachers would remind the children to tidy up their toys and clean their table, take their lunch bowls and spoons from their bags, wash their hands, line up to receive food, and return to their seats carefully. Sometimes, children fail to tidy up their toys and clean their tables, are unable to find their lunch bags, push others when lining up, and make noise in class. During mealtime, the preschool teachers must monitor the children’s eating to ensure they have eaten all the food, not only eating their preferred food, holding their lunch bowls and spoons correctly, and remain in their seats. When children are finished eating, the teachers would remind them to clean their lunch bowls, spoons, teeth, hands, and table. Children who demonstrate inappropriate behaviors delay the class routine, increasing the teachers’ stress. During mealtime, the teachers are concerned that the IMBs of children with disabilities may prevent them from obtaining proper nutrition but are also concerned that the children cannot participate in all scheduled class activities if they have spent too much time eating. Therefore, time management is critical for preschool teachers during mealtime.

The imitation of misbehaviors is a major challenge for preschool teachers. Children aged 3 to 6 years often imitate each other’s behaviors at this developmental stage. Therefore, when one child demonstrates inappropriate behaviors in class, other children may attempt to imitate those misbehaviors to obtain attention. If preschool teachers are unable to address these problem behaviors, such behaviors can quickly spread among children in the class, further disrupting class management. Therefore, when teachers observe inappropriate behaviors, they must immediately correct the misbehavior by introducing an alternative behavior.

The preschool administrator’s concern is another source of stress for preschool teachers. When children repeatedly exhibit problem behaviors in class, such as crying loudly, screaming, and attacking others, administrators may be concerned about the teacher’s ability to manage their class. The more behavioral problems that occur, the more concerns the administrator has about the teacher. The teacher may lose the support of the administrator, leaving the teacher in an isolated position. Because preschool administrators often monitor classes during lunchtime, the children’s mealtime behaviors are part of the administrator’s assessment of the teacher’s class management and problem-solving ability.

Although researchers have formulated strategies to improve IMBs, few were concerned about preschool teachers’ stress when dealing with these problem behaviors [7,8,9]. I speculated that the more stress preschool teachers experience, the less self-efficacy they have in teaching children with developmental disabilities. To understand how preschool teachers feel about children mealtime behavior problems, this study applied qualitative research to analyze the stress triggers of five preschool teachers who had intervened to address IMBs of children with developmental disabilities. The interview questions included the following:What mealtime problem behavior of children with disabilities in the class have you observed in class, and how did you deal with these problem behaviors?How did you felt when you were dealing with the children’s IMBs?What are the factors you think might influence children’s mealtime behaviors?

## 2. Literature Review

Children require sufficient nutrition to grow and remain healthy. For children with developmental disabilities, nutrition can somewhat compensate for their developmental disabilities and provide them with the energy required to participate in all preschool activities [10]. Sufficient and balanced nutrition assists children with developmental disabilities in participating in the same activities as children without disabilities in an inclusive learning environment [5,6,11]. However, mealtime behavior problems seriously impede the nutrition intake of children with developmental disabilities. Therefore, assisting these children in eating correctly enables them to obtain sufficient and balanced nutrition from food. Furthermore, sufficient energy and health can benefit the cognition of children with developmental disabilities, because playing and learning with children without disabilities increases the self-esteem, self-confidence, and personal value of children with developmental disabilities [12].

Emphasizing the critical nature of the nutrition of children with disabilities, the Individuals with Disabilities Education Act, Part H (1990) [5], regulates that schools or parents must provide nutrition that meets the children’s needs. Parents should be committed to supporting their children’s development and learning, respecting individual differences, and promoting their children’s self-awareness, competence, self-worth, resiliency, and physical well-being [11]. Article 1 of Taiwan’s Protection of Children and Youths Welfare and Rights Act (2021) [6] also regulates that schools and parents must promote the healthy development of children’s bodies and minds, protect their interests, and increase their welfare. Although the development and health of children in schools have been emphasized in different countries, the nutrition of children with developmental disabilities in preschools in Taiwan has not been explored. Research examining how schools can be supported to ensure children with and without disabilities have their nutritional needs met in preschools is lacking.

Children with, and without, disabilities often do not obtain sufficient nutrients [13]; many experience eating problems during mealtime at home or in school. Researchers have conducted studies and large-scale surveys and have reported parental concerns about their children’s problem behaviors at different ages. For example, parents of children aged between 6 weeks and 4.5 years primarily worried about their children’s sleeping, eating, and crying behaviors [13]. For children aged 1 and 2 years, parents were mostly concerned about their eating and sleeping difficulties. The number and intensity of parental concerns peaked when their children were aged 3 years and were related to difficulties with management and discipline. Other researchers reported that toileting, eating habits, and sleeping problems are common concerns for parents of 3-year-old children [14,15]. Because the development of children with developmental disabilities lags considerably relative to children without disabilities, the frequency, forms, and severity of their feeding problem behaviors are more apparent and more difficult to solve.

Secrist-Mertz et al. [16] argued that feeding problems for children with disabilities are complex and depend on the physical characteristics and nutritional needs of the child, the interactions between the child and feeders, and the child’s eating behaviors. The child’s physical condition may influence whether the child can successfully eat food. Children who have difficulty chewing and swallowing food, have particular feeding approaches, require dietary modifications, have low activity levels, or are treated using medications, may not have adequate nutrient intake. Parent-child interactions also influence children’s eating behaviors and may be affected by unique caregiving demands, particular feeding preferences, and pressure to eat [17]. Additionally, when the child demonstrates problem behaviors such as refusing to eat and having inflexible food preferences, ensuring the child receives adequate nutrition is difficult.

Common mealtime behavior problems of children with developmental disabilities include a low level of independent eating, food refusal, inflexible food preferences, and distraction from eating. Some children exhibit aggressive problem behaviors when they refuse to eat, including spitting food, pushing the table, breaking the chair, and other destructive behaviors. When children with developmental disabilities exhibit challenging behaviors, these problem behaviors become a source of stress for preschool teachers.

Alvarez [18] and Stormont [19] have observed that stressed teachers spend more than 20% of their class time engaged in negative interactions with children who exhibit problem behaviors, and only 5% of their class time in positive interactions with children. The USA National Prekindergarten study of 2003–2004 also indicated that 10.4% of state-funded prekindergarten teachers expelled at least one preschool child from their program, which was 3.2 times higher than that of children of other school grades. Amstad and Müller [20] noted that problem behaviors of children with disabilities are a source of stress for teachers, with kicking, hitting, and biting behaviors rated as the most stressful for teachers. Disruptive and antisocial behaviors were also reported as the most stressful behaviors in class [21]. Gebbie et al. [22] investigated the needs of preschool teachers in a North Carolina county in the United States. They noted that most teachers requested training on managing children’s challenging behaviors; those teachers who received training and mentoring in classroom behavior management strategies felt competent in managing challenging behaviors and increased their self-efficacy. When teachers can successfully solve problem behaviors, their stress is reduced.

Researchers have argued that Applied Behavioral Analysis strategies and techniques could successfully alleviate problem behaviors of children with disabilities, especially IMBs. ABA strategies include negative reinforcement, positive reinforcement, and escape extinction, which can assist teachers in modifying children’s mealtime behaviors [7,8,9]. Ahearn et al. [7] used negative reinforcement contingencies to physically guide children to accept food, whereby the feeder did not remove the spoon until the child accepted the presented food. Through this method, the children’s food acceptance increased. Hoch et al. [9] used positive reinforcement procedures and contingency contracting strategies and successfully increased children’s food acceptance. However, the children’s negative vocalization and class disruption behaviors did not change. Cooper et al. [8] also reported that using positive reinforcers increased the number of bites of food children took. Researchers revealed that reinforcement and escape extinction methods improved mealtime behavior problems in children with disabilities. The feeder used reinforcers to increase children’s appropriate eating behaviors and escape extinction to avoid children distancing themselves from the food they did not want to eat during mealtime [23,24,25].

Other educators claimed that the mealtime period presents an excellent opportunity to train children in positive social skills by teaching new techniques or replacement behaviors. Lalli et al. [26] used a behavioral consultation approach to reduce children’s problem behaviors and to increase their appropriate verbal behaviors during mealtime. Gaverea and Schwartz [27] used “snack talk” cards to increase appropriate social interaction among children with and without disabilities during mealtime. Therefore, I proposed that teaching appropriate techniques or behaviors successfully reduces children’s problem behaviors and increased proper behaviors during mealtime. As described in related research, children’s challenging behaviors are a source of teachers’ stress. The present study was conducted to determine how teachers feel when they are faced with children’s IMBs, the support they require, and the factors they believe are related to children’s mealtime behavior problems.

## 3. Research Method

### 3.1. Research Method

Semi-structured interviews were conducted with five preschool teachers who had experience with managing the IMBs of children with developmental disabilities in separate classes. Semi-structured interviews can provide rich and detailed perspectives on such children’s problem behaviors and how the teacher feels when dealing with those problems. Semi-structured interviews involve open-ended questions that allow teachers to provide in-depth and spontaneous responses [28]. This research approach was appropriate for collecting in-depth information to examine preschool teachers’ stress, problem-solving skills, and desired support.

### 3.2. Subjects

The sample consisted of five preschool teachers who had experience with managing IMBs of children with developmental disabilities in a classroom setting. The participating teachers were recruited through snowball sampling and were referred by collaborative preschool principles. Each teacher attended one-on-one interviews that each lasted at least 2 h. The interviews were held in a private room in the interviewer’s preschool. If required, the teachers attended multiple interviews. A total of 15 h of interviews were recorded. The interviewees were sent the questions before the interviews and the transcripts after the interviews. The background information of the five teachers is presented in Table 1.

As reported in Table 1, all interviewees were currently working as preschool teachers. All interviewees graduated with a bachelor’s degree in education, early childhood care, or special education. The average age of the children who exhibited mealtime problems was 3.7 (range: 2–5) years. The mealtime behavior problems included eating only preferred foods, eating only rice, eating only white toast, becoming distracted from eating, and not using eating utensils properly. All children had been identified as having developmental disabilities.

### 3.3. Interview Procedure

During a 2-h session, I trained a bachelor-level clinician in interview techniques to enable them to conduct the teacher interviews. I developed the interview questions, and the interviewer was informed of the study purpose and research design before the interviews.

The interviewer applied a semi-structured interview protocol, which outlined essential topics to generate rapport, gather information, and close the interview [29,30]. In addition, this study investigated the teachers’ problem-solving skills, stress triggers, effects, and factors influencing the effects. I used open-ended questions to facilitate in-depth discussion; if the discussion on a subject was incomplete, follow-up questions were used to collect additional information. I received all interviewees’ consent before audio-recording the interviews.

### 3.4. Interview Credibility

Epoché is crucial in qualitative research to prevent researchers’ preconceived ideas and beliefs about the phenomenon under investigation from biasing the results. To avoid personal beliefs influencing the outcomes, I trained the interviewer to conduct the interviews and developed the credibility process for this study.

Three primary methods were used to establish the credibility of the interviews. First, I examined the transcripts to ensure the interview content was relevant to the purpose of this study. Second, I conducted a triangulation process with two independent experts to reach a consensus on the themes extracted from the interviews that could be further analyzed. Third, I provided the interviewees with the interview transcripts to confirm their content.

### 3.5. Data Analysis

The data analysis process involved coding and breaking the data into segments. First, I used inductive coding to transcribe the recordings into numbered statements. I then clustered and identified themes that reflected each interviewee’s experiences and feelings. I employed the horizontalization process to identify relevant horizons [31], removing repetitive and irrelevant statements. The initial list of horizons was verified by comparing the interviews in sequence. I used the strategy of bracketing the data when reviewing the interview transcripts, in which the interviewees’ experiences were “bracketed” by adopting an “outsider” perspective and focusing on data that could be examples of the research topic. Finally, I compared the textural descriptions to develop a group composite textural descriptions to examine the phenomenon of teacher stress in the context of addressing children’s mealtime behavior problems in class.

## 4. Findings

The analysis of the semi-structured interviews revealed the following four major themes related to the interviewees’ experiences and feelings about addressing IMBs of children with developmental disabilities: (a) the identification of children’s mealtime behavior problems, (b) the application of problem-solving skills to address such problems, (c) the stress of solving such problems, and (d) the factors influencing children’s IMBs. In addition to sharing their experiences, reactions, strategies, and feelings when managing children’s mealtime behavior problems, the teachers also provided suggestions for improving the IMBs of children with developmental disabilities.

### 4.1. Theme 1: Identification of Children’s Mealtime Problems: “Different from and More Complicated Than How I Imagined.”

The preschool teachers had not expected that addressing the IMBs of children with developmental disabilities would be much different from, and more complicated than, addressing those of children without disabilities. Some teachers indicated that each child with a developmental disability had distinct eating problems during mealtime.


*“Andy has a problem with picking out his preferred food. If he sees something he does not like in the lunch bowl, he gets angry, cries, and screams in class. He even became so mad that he pushed the table away so the bowl and spoon fell to the floor.”*
(Teacher A: 50–52)


*“Bill only eats white rice. Although he does not eat much, he picks out the white rice to eat. After eating, he plays with his toy without noticing what’s happening in the classroom, including cleaning up. He does not eat breakfast if there is no white rice. He refuses to eat everything else, including dessert, if there is no white rice.”*
(Teacher B: 43–49)


*“Cody enjoys eating lunch and dessert in preschool but does not like vegetables. His major mealtime problem is that he uses his hands to grab food instead of using a spoon, and sometimes he eats while putting one foot on the chair, leaves his seat without permission, or drops rice on the table. He would also use his dirty hand to touch everything near him, and he would get distracted from eating. He throws the food he does not like on the table or leaves it in the bowl.”*
(Teacher C: 29–41)


*“David has a severe problem with picking out his preferred food. He only likes toast with strawberry jam and white rice with meat floss topping. When he comes across food he does not like, he will stop eating, leave the lunch on the table, or spit out the food. He also has a distraction problem; he becomes distracted from eating when other children are chatting or doing other things after they have completed the clean-up routine during mealtime. He also does not like chewing food when he is eating.”*
(Teacher D: 28–47)


*“Emily likes to eat meat or fried meat. She eats fast when she finds food she likes, but she picks food out if she does not like it. Moreover, she uses her bare hands to grab things she wants to eat and throws food away or on the floor if she does not like it. Sometimes, she just sits still refusing to eat, or spits out the food she does not like. She also gets distracted from eating during mealtime. Another problem is using a spoon. She cannot hold a spoon correctly, so she sometimes drops rice on the table or floor.”*
(Teacher E: 38–69)

According to the interviews, children’s IMBs include picking out preferred foods, refusing to eat, using hands instead of a spoon to grab food from the lunch bowl, putting one foot on the chair when eating, dropping rice on the table or floor, becoming distracted from eating, and purposefully spitting out food. Picking out preferred food is the most common problem with mealtime behavior of children with developmental disabilities. Children may demonstrate various misbehaviors when they encounter food they do not enjoy, such as screaming, crying, pushing the food away, picking out food, refusing to move, vomiting, and leaving the table. The teachers observed that IMBs are more severe and complex in children with developmental disabilities than in children without disabilities.

### 4.2. Theme 2: Problem-Solving Skills: Persuading or Direct Teaching?

When faced with children’s IMBs, most preschool teachers preferred not to deal with the problem behaviors immediately when the child demonstrating such behaviors is in a non-compliant mood. If the child was not angry or violent, some of the teachers would intervene directly and teach the child the appropriate behaviors. Some attempted to communicate with the child first before applying any strategy.


*“When Andy sees food he does not like in his bowl, he gets angry. First, I help him pick out the food he does not like and put it on the bowl cover. Then, I try to talk to him to slowly persuade him to accept the food. It takes a long time, but I will not force him to eat everything.”*
(Teacher A: 50–64)


*“Bill eats only white rice, and when he is finished eating, he plays with his toys without following the class routine. I tell other children in the class that he can play with his toy because he has finished eating. Because this situation has been ongoing for a while, I know he likes a particular car. So, I put the car he likes in front of his lunch and tell him, ‘If you can try a little bit of food other than the white rice, I will let you play with the car.’”*
(Teacher B: 102–107)


*“Cody has a variety of behavioral problems. I applied for a special education teaching assistant. This teaching assistant and I are jointly responsible during mealtime. For example, I help prepare the children’s meals, and the assistant helps the children to get ready to line up for lunch. The teaching assistant sets the IEP objectives to help the child learn to have lunch while seated, use a spoon correctly, and clean up after mealtime. When he is finished, he is allowed to play with the children. I also use related picture books to teach the children which utensils we use to eat, and we all discuss them in class. Then, I use the pictures to remind the children to eat with proper manners.”*
(Teacher C:237–272)


*“David has a problem with picking out his preferred foods, but I insist he finishes the food in the bowl without running away. I encourage him to go to the corner of the room he likes if he finishes the meal. If he does not finish the meal, he is not allowed to play in his favorite corner. I continually remind him to eat without picking out his preferred food and to improve his communication skills. I believe that each child has individual differences. If we give children more time, they can do well.”*
(Teacher D: 58–75)


*“Because Emily has a problem with becoming distracted from eating, she needs to learn and practice concentration. I have to watch her eating all the time during mealtime. When she tries to use her hand to grab food, I tell her to use a spoon to eat. There are two teachers trying to fix her mealtime problems. One teacher asks her to finish the food in the bowl, even if it is only one bite of food left. The other teacher does not have time to watch Emily constantly during mealtime, so if Emily does not finish the meal, the teacher feeds her. When Emily spits out the food, the teacher simply gives her another spoonful so she understands that spitting out the food is no use. We also ask her to finish all her food before brushing her teeth, because some children keep the food in their mouths without swallowing it.”*
(Teacher E: 82–94)

The preschool teachers applied different techniques to persuade or teach the children to eat appropriately, such as discussion, feeding, and the use of reinforcement procedures. Teacher B used Bill’s favorite car toy as a reinforcer to encourage him to eat, and Teacher D used David’s favorite playing corner to encourage him to improve his mealtime behavior problems. Some of the teachers applied the extinction strategy to prevent children from escaping from their meals, such as Teacher D and Teacher E. Teacher C used a more systematic procedure to teach Cody correct mealtime behaviors. For example, the special education teacher developed IEP goals, and Teacher C and the assistant collaboratively assisted Cody in achieving these goals.

### 4.3. Theme 3: The Stress of Solving Problems: Powerless and Self-Doubt

When dealing with the children’s IMBs, the preschool teachers tended to feel powerless and self-doubting, because they may not see the positive and successful outcome of the intervention. Furthermore, they must also deal with the children’s other problem behaviors during class time. The preschool teachers described the characteristics of each child with special needs, revealing their stress and concerns from managing these children’s problem behaviors.


*“Andy cannot express his emotions. Therefore, when he is doing an activity and is interrupted, he becomes angry. He only does activities he likes, and I feel very stressed when changing activities. There are six children with special needs in the school, and we have three children with special needs in my class but only two teachers. I teach Andy when he listens to me, but I cannot do anything if Andy is so angry that he refuses to listen.”*
(Teacher A: 23–48)


*“Most private preschools do not accept children with disabilities. Two young children in my class were diagnosed after they enrolled. I feel a little stressed because, at school, I was training Bill to feed himself without picking out food, but his grandmother usually feeds him at home and tells me nothing about his food or eating behaviors. Therefore, he does whatever he wants in class, and it is difficult for me to teach him proper mealtime behaviors.”*
(Teacher B: 77–96)


*“So, the assistant and I help Cody with many tasks that he should complete during mealtime. Because if we do not help him, he will disrupt the whole class schedule. Besides, I also worry that his bad eating behaviors might make him sick during mealtime. Cody does not live with his parents; he is an aboriginal resident of a rural area and lives with his aunt, and his parents work in another city. Therefore, it is challenging to meet and communicate with his parents. I have to teach Cody all behaviors in school.”*
(Teacher C: 48–58)


*“Emily has a developmental disability that affects her speech and communication, and she has a problem with self-care and emotional stability. She has undergone speech therapy. Because she spits out the food on the floor when the other children are eating, I ask the other children to leave and clean it up. She also has a problem with concentrating on tasks. When we ask her to do something, she looks at what the other children are doing and forgets to do what she was told. Therefore, I feel stressed when I must constantly watch and remind her what she should be doing. I also feel stressed when parents do not cooperate with the teacher. Although I teach Emily to eat everything in her bowl without picking out food, her parents allow her to pick out the food she does not like. Inevitably, she becomes angry and demonstrates inappropriate mealtime behaviors when she cannot do whatever she wants in school.”*
(Teacher E: 82–101)

The preschool teachers’ stress resulted primarily from the children’s problem behaviors and a lack of support for the teachers from the children’s families. The problem behaviors of the children have exceeded the scope of the preschool teachers’ understanding, and parents’ inconsistent discipline and laissez-faire attitude further negatively affect the results of the teachers’ interventions in school.

### 4.4. Theme 4: Factors Influencing Children’s Mealtime Behaviors: Family Support Is the Key

The preschool teachers believed that several factors affect the children’s IMBs, including health problems, emotional status, motor development, disabilities, parental socioeconomic status, and inconsistent discipline. However, some of the preschool teachers indicated that family support is critical in addressing and educating children on mealtime behaviors.


*“When Andy was sick, he ate less. He takes longer than the other children to eat during mealtime, and I have to try different methods to improve his mealtime behaviors. When he can express his emotion, his is less aggressive.”*
(Teacher A: 55–73)


*“His problem with picking out food was so serious when I first met him, and he was still sucking on the pacifier at 2 years old. His grandparents let him do whatever he wants, and I believe that his mealtime behavior problems are related to his eating habits at home.”*
(Teacher B: 109–144)


*“Cody was diagnosed with a developmental disability and was not the only one with a disability in the class. Therefore, I feel powerless when I have to deal with many similar situations at the same time. If other children have ADHD problems, managing the class is more challenging. Cody’s parents are of a low socioeconomic status and have to work outside the village to earn a living to support their family. I believe the characteristics of Cody’s disability and his family support are primary factors influencing his mealtime behaviors.”*
(Teacher C: 154–169)


*“David has a developmental disability affecting his cognitive and language abilities, but he can understand simple commands. He does not like to chew and does not like to eat when he feels sick.”*
(Teacher D: 15–38)


*“I think different disciplinary methods influence each child’s mealtime behaviors. I ask Emily to finish what she has in her lunch bowl, but Emily can decide the amount she wants to eat. However, when another teacher uses different disciplinary methods to address Emily’s inappropriate mealtime behaviors, such as feeding her, I found Emily’s problem behavior is worse than before.”*
(Teacher E: 72–79)

Some of the preschool teachers believed that IMBs of children with developmental disabilities are attributable to their developmental limitations, and others maintained that such behaviors are influenced by the children’s family support.

## 5. Discussion

The purpose of the interviews was to explore the experience of five preschool teachers with experience in addressing IMBs of children with developmental disabilities in class. The research focused on identifying the main IMBs of children with developmental disabilities, teachers’ strategies or levels of support for managing such behaviors, teachers’ stress, and factors contributing to children’s IMBs. Four major themes emerged during the study: (a) the identification of children’s mealtime behavior problems, (b) the application of problem-solving skills to address such problems, (c) the stress of solving such problems, and (d) the factors influencing children’s IMBs. As the number of children with developmental disabilities enrolling in inclusive preschool classrooms increases, the need for preschool teachers to understand the children’s individual needs and characteristics also grows. When addressing IMBs, teachers experienced stress through feelings of uncertainty, isolation, and powerlessness. Because the problem behaviors of children with developmental disabilities are different from those without disabilities, identifying an effective intervention strategy to improve children’s behaviors is difficult for teachers. Even with the extra support provided in Teacher C’s case, where a teaching assistant assisted in improving children’s IMBs, the outcome was still limited. Therefore, what form of support can be helpful and practical for solving IMBs of children with developmental disabilities in preschools must be determined.

Because children with developmental disabilities demonstrate mealtime behavior problems that are more difficult and complex than children without disabilities, addressing these behavioral problems through discussions or commands only may be ineffective. The form, intensity, and frequency of IMBs of children with developmental disabilities differ from those of children without disabilities and can generate more stress for teachers

This study has several limitations, which must be considered before generalizing the findings to other preschool teachers. First, the sample was small, representing the views of only five preschool teachers; they thus may not reflect most preschool teachers’ experiences in inclusive classrooms. Therefore, more preschool teachers must be recruited in further studies to investigate the IMBs of children with developmental disabilities by using a quantitative research method or mixed-method approach to establish reliable information. Second, the findings of this study were obtained primarily from preschool teacher interviews; thus, this study lacked additional data resources, such as observation records of children’s behaviors, IEPs, and classroom management notes. Future research is required to further examine the effectiveness of the support and strategies provided by preschools. Such research could assist preschools in developing well-designed and practical supportive models to reduce preschool teachers’ stress. Furthermore, future studies could focus on successful interventions for addressing IMBs of children with developmental disabilities to increase preschool teachers’ self-advocacy and advocacy of inclusive classes.

## 6. Recommendations for Future Preschools

Preschools should implement measures to reduce the stress of teachers managing IMBs of children with disabilities. First, preschools can assign a mobile support teacher to classrooms in which support is required immediately. Second, preschools must hire experienced professionals to effectively address children’s problem behaviors in class. Third, to increase intervention success, preschools can develop a support system to assist teachers in determining when and for whom interventions or strategies are implemented, what the intervention or strategy is, and how the intervention or strategy can be continually evaluated and modified as necessary. Fourth, communication with children’s families is critical for influencing children’s eating behaviors. Therefore, preschool teachers must learn to effectively communicate with the children’s families to gain the families’ support. Finally, preschool teachers must regularly communicate with preschool principals or directors to ensure that the principals or directors understand the preschool teachers’ stress and the difficulties of managing children’s behavior problems.

The findings of this study indicate that many preschool teachers do not know how to quickly and effectively solve the mealtime behavior problems of children with developmental disabilities, which increases teachers’ stress. Therefore, the practical approaches taught in higher education must focus on teaching teachers how to identify, manage, and change children’s problematic behaviors to ensure that the teachers are well-prepared for classroom settings.

The IMBs of children with developmental disabilities cannot be ignored because they directly relate to the children’s health, development, and educational activities in preschool. Developing an effective support system for preschool teachers to improve their skills in solving children’s mealtime behavior problems is imperative. This research provided valuable insight into preschool teachers’ experiences when encountering mealtime behavior problems of children with developmental disabilities, and the results provide a starting point for examining the curriculum of higher education teacher training and the need to provide preschool teachers with adequate class management skills.

## Figures and Tables

**Table 1 children-10-00441-t001:** The Background Information of the Interviewees.

Teacher	Age	Major	Seniority	Children’s Name	Children’s Age	Children’s Mealtime Problem
A	35	Bachler’s in Education	9 years	Andy	5	Selecting food.
B	28	Master in dept. of social and local development	6 years	Bill	2	Eat rice only, eating distractions.
C	28	Master’s in special education	4 years	Cody	4.5	Eating without using spoons properly, eating distractions.
D	40	Bachelor’s in early childhood care and education	8 years	David	8	Eat white toast only.
E	22	Bachelor’s in Early childhood care and education	4 years	Emily	4	Selecting food.

## Data Availability

I confirmed that the data supporting the findings of this study are available within the article. The descriptive data will be available upon reasonable request.

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
