# Peer review of "The Mealtime Behavior Problems of Children with Developmental Disabilities and the Teacher’s Stress in Inclusive Preschools"

_children, 2023, doi:10.3390/children10030441_

Round 1

Reviewer 1 Report

A useful manuscript on an important topic.

Title is poorly written and inaccurate.

A more accurate title would be: Teacher stress due to mealtime behavior problems of children with developmental disabilities in inclusive preschools

The first two sections are poorly structured in terms of their content developing a logical narrative. There are also many grammatical errors throughout the manuscript. For example:

Line 10: and experiencing malnutrition

Line 14: and maintain (not maintains)

Line 20 & 21: selecting specific the feeder – this phrase does not make sense

Line 33: disabilities occur inappropriate – this phrase does not make sense

Line 47: would occur problems – this phrase does not make any sense

The information presented in Table 1 suggests that each teacher was interviewed with reference to the behaviour of one specific child. This is an important feature of the methodology which is not mentioned in the text of the manuscript. I assume that consistent with good ethical practice in research, the names used for the children in Table 1 are not their true names? This should be clearly stated in the manuscript

The average data presented at the top of page 6 is unnecessary for a qualitative study based on interviews of five subjects.

The narrative switches between “I” and “we”. Yet the manuscript only has one author

Typesetting error: First paragraph under 4.2 should not be in italics

Author Response

Reviewer’s Feedback Responses

The manuscript has been grammatical edited

Reviewer 1

Reviewers Feedback

Responses

Title is poorly written and inaccurate.

A more accurate title would be: Teacher stress due to mealtime behavior problems of children with developmental disabilities in inclusive preschools

Title has been modified to: Teacher stress due to mealtime behavior problems of children with developmental disabilities in inclusive preschools

The first two sections are poorly structured in terms of their content developing a logical narrative.

To develop a logical narrative, I have deleted the first graph.

There are also many grammatical errors throughout the manuscript.

This manuscript was edited by Wallace Academic Editing.

Line 10: and experiencing malnutrition

This paragraph has been deleted because it is not directly related to the content.

Line 14: and maintain (not maintains)

“it helps them develop and maintains energy to go through all activities in school.”

This paragraph has been deleted because it is not directly related to the content.

Line 20 & 21: selecting specific the feeder – this phrase does not make sense

Children with disabilities used to choose a specific feeder during the mealtime. For example, a child with autism would eat with Teacher A besides him instead of Teacher B. This is a unique problem during the mealtime time.

Line 33: disabilities occur inappropriate – this phrase does not make sense

Line 51“many children with developmental disabilities occur inappropriate mealtime behaviors in preschools, which may result in their lack of nutrition intake.”

Line 45 “I observed that many children with developmental disabilities demonstrate IMBs in preschools, which may result in an insufficient nutritional intake.”

Line 47: would occur problems – this phrase does not make any sense

Line 64 “Children at this time would occur problems, such as eating distractions, selecting food, eating too quickly or too slowly, eating without chewing, cannot eat independently, or cannot use spoons appropriately.”

After reviewing the sentence, I also agreed the reviewer’s comment and deleted the sentence.

The information presented in Table 1 suggests that each teacher was interviewed with reference to the behaviour of one specific child. This is an important feature of the methodology which is not mentioned in the text of the manuscript. I assume that consistent with good ethical practice in research, the names used for the children in Table 1 are not their true names? This should be clearly stated in the manuscript

I have added line 234-237 with related information. I also added detailed IRB committee information to the attachment of this manuscript. “All children’s background information that the teachers mentioned were fully protected and anonymous. I collected all the information from the teachers and nick-named their target children to protect their privacy. The Research Ethics Review Committee has proofed this study at National Tsing Hua University.”

The average data presented at the top of page 6 is unnecessary for a qualitative study based on interviews of five subjects.

I have deleted the average data.

The narrative switches between “I” and “we”. Yet the manuscript only has one author

I have modified all “we” into “I” consistently.

Typesetting error: First paragraph under 4.2 should not be in italics

I have corrected.

Reviewer 2 Report

This report deals with the difficulties teachers face in mealtime behavior in children without developmental disability.

Nutrition and feeding problems in children with developmental delays is a topic of interest, often neglected by aspects such as stimulation in cognitive, motor or communicative areas in the framework of early intervention.

For this reason, reports such as this one focus on the school context, which, as the author points out, is also strongly influenced by the family context.

From a formal point of view, the writing and editing are adequate, with the exception of some quotations that do not seem to follow the format of the others (122, 126 and 129) and the use of italics in 343-346.

Regarding the content, I have some considerations to make to the author:

First, I find that some of the information is repetitive, as children's feeding problems are described in different parts of the paper. The information collected in 160 is already so in the introduction.

Complementarily, I miss some reflection on children's feeding problems with a normative development on the one hand and, most importantly, teachers' stress.

Regarding the first of these issues, 469-470 does not describe for us whether normatively developing children show any difficulties in mealtime and if so, what is the role of the family. We may think that the underlying process (allowing the child to eat what he/she wants and in the way he/she wants) is the same although the problems are undoubtedly of greater magnitude in the case of children with disabilities. It may also be that they are different processes and that parental overprotection operates differently among children with disabilities, but such reflection is advisable. 

However, my major objection is around the paper's treatment of teacher stress: In a certain way, the title of the article is misleading as it dwells much more on children's problems than on teacher stress. Little space is devoted to the latter. It is a measurable construct of which we do not know its prior levels in the sample at the time of the study. In other words, are teachers comparable in stress regardless of whether their students with developmental difficulties present behavioral problems, or is stress generated only in this context?

The analysis of the results and the derived conclusions seem appropriate to me, bearing in mind the above objections. In the limitations of the study, the author himself points out two of these: the small sample and the need to extend the study from the methodological point of view.  In the case of introducing a quantitative perspective, stress would become the DV and would require a theoretical framework in which it would occupy a central place, which is not present in the current work. 

Therefore, my recommendations are aimed at the clarification of the aspects mentioned above and the reflection on the title, which should focus on food problems and not on teacher stress, since in that case the whole introduction and state of the art would be incomplete without  focussing on stress.

Author Response

Reviewer’s Feedback Responses #2

The manuscript has been grammatically edited

Reviewer 2

Reviewers Feedback

Responses

First, I find that some of the information is repetitive, as children's feeding problems are described in different parts of the paper. The information collected in 160 is already so in the introduction.

The prior information is about the feeding problems of children with developmental disabilities, and the feeding problems described in 160 primarily focused on what the teacher feels are the most stressful mealtime problems of children in the class. Therefore, the readers understand what stressful problem behaviors for the teacher in class are.

Complementarily, I miss some reflection on children's feeding problems with a normative development on the one hand and, most importantly, teachers' stress.

1.          Since the title is focused on the feeding problems of children with disabilities, I would not particularly mention the problems of children with normative development in this paper.

2.          The teacher’s stress came from the feeding problems of children with disabilities in class. I have collected the teacher’s reflections on dealing with feeding problems of children with disabilities on theme 3 “the stress of solving problems: powerless and self-doubt.” In addition, the discussion part also mentioned what the teacher’s stress came from and what the recommendations are.

3.          As I reviewed the paper, I believed the primarily part of this paper still focused on the mealtime problem behaviors of children with disabilities in class and less discussed the teacher’s stress. To meet the content of the paper, the title has changed to “The mealtime behavior problems of children with developmental disabilities and the teacher’s stress in inclusive preschools.”

Regarding the first of these issues, 469-470 does not describe for us whether normatively developing children show any difficulties in mealtime and if so, what is the role of the family. We may think that the underlying process (allowing the child to eat what he/she wants and in the way he/she wants) is the same although the problems are undoubtedly of greater magnitude in the case of children with disabilities. It may also be that they are different processes and that parental overprotection operates differently among children with disabilities, but such reflection is advisable. 

1.          Since this paper mainly discussed the feeding problems of children with disabilities in class, I would not talk about the feeding problems of children with normatively developing children.

2.          Although some children with normative development have feeding problems in a preschool class, the teachers might not have to spend much more time one-on-one to deal with those problems. In Taiwan’s preschools, most children with normative development can adapt to mealtime behaviors and routines in class under the teacher’s management.

3.          Thank you for the reviewer’s advice, and I will consider discussing this topic in the future.

My major objection is around the paper’s treatment of teacher stress: In a certain way, the title of the article is misleading as it dwells much more on children’s problems than on teacher stress. Little space is devoted to the latter. It is a measurable construct of which we do not know its prior levels in the sample at the time of the study. In other words, are teachers comparable in stress regardless of whether their students with developmental difficulties present behavioral problems, or is stress generated only in this context?

1.          To review this paper, I also think the title might change to the mealtime problem behaviors of children with development disabilities rather than teacher stress. So, the title has changed to “The mealtime problem behaviors of children with developmental disabilities and the teacher’s stress in inclusive preschools.”

2.          The reason to focus primarily on the mealtime problems of children with developmental disabilities rather than others is that mealtime behaviors and routine are the important concern of children with disabilities and their families. Most of our preschool children spend more than 6 hours in school, 6 hours in public preschools, and more than 8 hours in private preschools. In other words, school meals probably would be their major nutrition source. Furthermore, some children with disabilities also come from poor families. Their families might have difficulties providing enough food for them. Due to the importance of nutrition intake for children with developmental disabilities, it is important to ensure they obtain enough food and nutrition. That is the primary issue for this paper. Therefore, I only focused on the mealtime problem behaviors of children with disabilities and their teacher’s stress in class other than the other problems issues.

The analysis of the results and the derived conclusions seem appropriate to me, bearing in mind the above objections. In the limitations of the study, the author himself points out two of these: the small sample and the need to extend the study from the methodological point of view.  In the case of introducing a quantitative perspective, stress would become the DV and would require a theoretical framework in which it would occupy a central place, which is not present in the current work. 

1.          As the title changed to the mealtime problem behaviors and the teacher’s stress of this paper. The DV would consider the mealtime problem behaviors of children with disabilities and the teacher’s stress, and the IV would be the intervention, which is how the teachers solved those problems.

2.          Therefore, the theoretical framework would begin with the nutrition and feeding problems of children with disabilities to the teacher’s stress in class.

My recommendations are aimed at the clarification of the aspects mentioned above and the reflection on the title, which should focus on food problems and not on teacher stress, since in that case the whole introduction and state of the art would be incomplete without focusing on stress.

1.          I agree the reviewer’s point. I have changed the title to focus primarily on the mealtime problem behaviors of children with disabilities rather than the teacher’s stress. However, I also believe that solving mealtime problem behaviors also caused the teacher stress. Therefore, I would first focus on the mealtime problem behaviors of children with developmental disabilities and also mention how the teachers feel when dealing with those problems in class.

Round 2

Reviewer 1 Report

The changes have resulted in a significant improvement. The changed title is not as reported by the author, but is acceptable.

Reviewer 2 Report

After carefully reading the authors' responses, I consider the assessents made in the first round of this evaluation to be sufficiently justified. The change in the title also responds to the suggestions of this reviewer.